# Trend of Ultraprocessed Product Intake Is Associated with the Double Burden of Malnutrition in Mexican Children and Adolescents

**DOI:** 10.3390/nu14204347

**Published:** 2022-10-17

**Authors:** Cecilia Isabel Oviedo-Solís, Eric A. Monterrubio-Flores, Gustavo Cediel, Edgar Denova-Gutiérrez, Simón Barquera

**Affiliations:** 1Center for Nutrition and Health Research, National Institute of Public Health, Cuernavaca 62100, Morelos, Mexico; 2School of Nutrition and Dietetics, University of Antioquia, Medellín 050010, Colombia

**Keywords:** double burden of malnutrition, anemia, obesity, ultraprocessed products, children, adolescents, Mexico

## Abstract

Background: Ultraprocessed products (UPPs) have been associated with unfavorable health outcomes; however, until now, they have not been associated with the coexistence of undernutrition and overnutrition, known as the double burden of malnutrition (DBM) at the individual level. Methods: Cross-sectional analyses were performed on data collected from children and adolescents participating in the 2006 and 2016 Mexican National Health and Nutrition Surveys. The food and beverages reported in the food frequency questionnaire (SFFQ) were classified as UPPs as defined by the NOVA classification system. Associations of UPPs with anemia, excess weight, and the DBM were estimated with logistic regression models. A pseudo-panel was generated using the cohorts of children born from 1997 to 2001 to estimate the effect of the UPPs on anemia, excess weight, and the DBM. Results: The consumption of UPPs (% energy) was higher in 2016 (children 30.1% and adolescents 28.3%) than in 2006 (children 27.3% and adolescents 23.0%) in both age-groups. The higher contribution of UPPs was positively associated with excess weight and the DBM in children’s lower tertile of socioeconomic status (SES) and the DBM in higher tertile of SES in adolescents. The pseudo-panel analysis shows the positive association between UPPs and DBM in lower SES. Conclusions: These results provide evidence of the association between the consumption of UPPs and the DBM and excess weight in children and adolescents.

## 1. Introduction

The definition of the double burden of malnutrition (DBM) spans individuals, households, and populations throughout a lifetime [1]. The DBM is the coexistence of undernutrition and overnutrition [2]. Environmental, social, cultural, economic, commercial, and political factors determine the double burden of malnutrition [3,4].

In 2016, in Mexico, more than 3 in 10 children (5 to 11 years old) and adolescents (12 to 19 years old) had excess weight (girls = 32.8%, and adolescent females = 39.2%; boys = 33.7%, and adolescent males = 33.5%) [5]. Moreover, the prevalence of anemia was 12.5% in children and 9.6% in adolescents in 2016 [6]. A previous study in 2012 showed that the prevalence of DBM (anemia and overweight) in children was 2.9% [7].

The occurrence of two forms of malnutrition may reflect the stage of the nutritional transition caused by reduced physical activity and increased access to less healthy, highly processed products and beverages labeled as ultraprocessed products (UPPs) [8,9]. This problem is driven by the food system and the obesogenic environment [10,11].

UPPs are industrial formulations of food substances and additives to create ready-to-eat or drink products. These products are hyperpalatable and have sophisticated and attractive packaging and aggressive marketing [12]. In general, UPPs dominate food supplies in middle- and low-income countries [13], displacing dietary patterns based on traditional, freshly prepared meals with the constant consumption of energy-dense, fatty, sugary, or salty ready-to-eat products [14,15]. Over the last three decades, purchases of UPPs have doubled in Mexico, while purchases of unprocessed or minimally processed foods have gradually decreased [16]. In addition, children and adolescents consume more UPPs compared to adults [17]. Consequently, UPP consumption reduces dietary diversity and micronutrient intake in the Mexican population [18]. Some epidemiological studies have shown that diets characterized by high consumption of UPPs have low nutritional quality [19]. Higher consumption of UPPs increases energy density, added sugar, total fat, and saturated fat, while the content of proteins, dietary fiber, dietary diversity, and micronutrients (e.g., niacin, pantothenic acid, pyridoxine, folate, vitamin B12, zinc, calcium, magnesium, potassium, and phosphorus) decreases [18,19,20].

For younger ages (children and adolescents), UPPs have been associated with unfavorable health outcomes [21,22,23]. These include excess body fat, abdominal obesity [24,25,26,27], high low-density lipoprotein cholesterol, low high-density lipoprotein cholesterol [28,29], and food addiction [30]. Additionally, there is an association between lower (vs. higher) consumption of UPPs and the reduction of diastolic blood pressure [31].

Despite the evidence on the adverse health effects of UPPs intake and the poor nutritional profile [19], to our knowledge, whether there is or is not an association between UPPs and two forms of malnutrition at the individual level has not been identified. Therefore, the current study examined the association between the consumption of UPPs and the coexistence of anemia and excess weight in Mexican children and adolescents in two nationally representative surveys.

## 2. Materials and Methods

### 2.1. Participants and Study Design

Cross-sectional analyses were performed on data collected from children and adolescents participating in the 2006 and 2016 Mexican National Health and Nutrition Surveys (ENSANUT, by its Spanish acronym). The ENSANUT is a nationally representative, probabilistic, and multistage stratified sampling survey. The details of the sample selection and its characteristics have been previously published [32,33].

The ENSANUT 2006 was conducted between October 2005 and May 2006 in 48,304 households and obtained information from 15,111 children (5 to 11 years old) and 14,578 adolescents (12 to 19 years old). The ENSANUT 2016 was conducted between May and September 2016 in 12,163 households and gathered information from 2584 children and 3830 adolescents [33].

The present analysis included information on a probabilistic subsample of children and adolescents who completed the dietary assessment and from whom we had blood sample information available to determine hemoglobin concentration, anthropometric measurements, and physiological status (nonpregnant and nonlactating). This study analyzed 8739 children and 7299 adolescents who were in the appropriate age span and eligible to be included in the analyses in 2006. In 2016, 3251 children and 2379 adolescents were analyzed. We excluded participants with potentially implausible dietary data [34]. We also excluded from the analysis those participants who had consumed more than three foods (grams) above three standard deviations (3 SD) and ratios above +3 SD of energy intake/estimated energy requirement (EER). The EER was estimated using equations from the Institute of Medicine [35]. Moreover, to clean data at the lowest extreme energy-intake values, we excluded participants with energy intake/basal metabolic rate ratios below 0.5 according to Food and Agriculture Organization equations [36]. Additionally, we excluded participants with implausible <−5 or >+5 body mass index (BMI) z score [37], and hemoglobin in children and adolescent females <40 g/L or >185 g/L, and adolescent males <40 g/L or >200 g/L, respectively [32]. Thus, the final study sample was 8074 children and 6482 adolescents from the 2006 survey and 2934 children and 2118 adolescents from the 2016 survey.

The present study was developed and performed according to the Declaration of Helsinki guidelines. Adults responsible for children and adolescents gave informed consent, and children gave informed assent. The Research, Biosafety, and Ethics Committees of the Mexican National Institute of Public Health approved the survey and the secondary analysis.

### 2.2. Dietary Assessment

Dietary intake information was obtained using the semiquantitative food frequency questionnaire (SFFQ) designed for ENSANUT. The SFFQ of the ENSANUT 2006 includes 101 different foods and beverages [38]. The SFFQ in 2016 was based on SFFQ 2006 and incorporated 39 additional food items selected by a group of experts in nutrition from the Center for Nutrition and Health Research, National Institute of Public Health, in Mexico [34]. The SFFQ in ENSANUT MC 2016 includes 140 foods and beverages [34]. Trained and standardized personnel and interviewers asked study participants or the person responsible for feeding a child under 12 years of age how frequently they consumed these foods and beverages seven days before the interview. The interviewers used the software Visual FoxPro program, v.7 (Microsoft, CA, USA), to input data [34,38].

For estimations, the number of days was multiplied by the number of times per day that the food and beverage item was consumed in the last seven days, and then the portion size per day was calculated. To calculate the energy consumption (kcal/d), we multiplied the daily frequency of consumption (portions/d) of each food and beverage by the energy content of the food and beverage using the food composition tables compiled by the Institute of Public Health in Mexico [39].

### 2.3. Ultraprocessed Product Definition

The foods and beverages reported in the SFFQ were classified as UPPs as defined within the NOVA classification system, which groups foods according to the extent and purpose of industrial processing [12]. The NOVA classification identifies, in addition to UPPs, three other groups defined as unprocessed and minimally processed foods (MPFs), processed culinary ingredients (PCIs), and processed products (PPs). Details of UPP classification and validation of SFFQ to estimate dietary intake according to the NOVA classification system were previously reported [40]. Appendix A shows all items corresponding to UPPs in the SFFQ.

### 2.4. Definitions of the Main Variables

Double burden of malnutrition: The DBM was defined as the coexistence of anemia and excess weight at the moment of the study measurements.

Anemia: Hemoglobin (Hb) was measured using a portable HemoCue Hb photometer (HemoCue, Angelholm, Sweden). The Hb concentrations to diagnose anemia (adjusted by altitude) [41] were <11.5 g/dL for children 5 to 11 years old, <12.0 g/dL for adolescents 12 to 14 years old, <12.0 g/dL for females ≥15 years old, and <13.0 g/dL for males ≥15 years old.

Excess weight: This was defined as overweight or obesity. The body mass index (BMI = kg/m^2^) z score for age and sex was calculated for children and adolescents according to the standard of reference of the World Health Organization (WHO) [42]. The z score was calculated using the software WHO AnthroPlus v 1.0.4 (Department of Nutrition World Health Organization, Geneva, Switzerland). Children and adolescents with z scores above +1 were classified as overweight, and those with z scores above +2 standard deviations as obese. Weight was measured using an electronic balance with an accuracy of 100 g and length with a stadiometer with an accuracy of 1 mm [32,37]. The procedures for measuring weight and height used for calculating BMI have been described elsewhere [43].

### 2.5. Sociodemographic Characteristics

Socioeconomic status (SES): SES was derived using the principal component method. The final model included information on housing characteristics and possession of goods. The first standardized index score was divided into tertiles representing low, medium, and high socioeconomic status.

Place of residence: Communities with fewer than 2500 residents were considered rural, and those with 2500 or more residents were considered urban.

Region: The country was divided into three regions: north, central (including the Mexico City Metropolitan Area), and south. The states included in each region were previously published [37].

Head of household education level: Head of a household education level was classified according to the maximum years of study completed: low—no formal education to elementary school; medium—middle and high school education; and high—college and postgraduate courses.

Indigenous status: The self-report on indigenous status came from the question “Do you consider yourself indigenous?” The answer was yes/no.

Screen time: This was defined as time spent watching television and videos or playing video games on TV or computers [44] and categorized as less than 7 h/week, 7 to 13 h/week, 14 to 20 h/week, and 21 or more h/week.

### 2.6. Statistical Analysis

Percentages with their respective 95% confidence intervals (95% CIs) were used to describe the prevalence of DBM, health conditions, and sociodemographic variables. Averages with their respective 95% CIs were used to describe total energy intake and the consumption of UPPs.

The two surveys were included in a logistic regression model to describe the association between DBM, anemia, and excess weight with UPPs in 2006 and 2012. Then the energy consumption of UPPs was presented for every 200 kcal. For all outcomes, we estimated crude (model 1) and adjusted (model 2) odds ratios (OR) and their 95% CI. The analysis stratified by SES was shown. Model 1 was adjusted only by survey year and energy NOVA groups (energy partition model) [45]. Model 2 was adjusted by variables model 1, age quintiles, SES, indigenous status, place of residence, and region, plus screen time and sex in adolescents. The variables adjusted in model 2 were considered potential confounders at the same time parsimonious model.

The pseudo-panel analysis estimates panel data models from a time series of independent cross sections. The approach proposed is to divide the population into a number of cohorts (these being groups of individuals sharing some common characteristics) and to treat the observed cohort means as error-ridden measurements of the population cohort means [46]. For the pseudo-panel analysis, restricted cells to the birth cohort from 1997 to 2001 were constructed for every country region, place of residence, SES, and sex. For every cell, the prevalence of DBM and health conditions were estimated. The deltas (change from 2006 to 2016) of the prevalence of DBM, health conditions, and consumption of UPPs and MPFs were calculated. We used a linear regression model using robust estimation to explain the DBM delta using UPPs delta adjusted by DBM conditions in 2006 and MPF delta and stratified by SES status. Statistical significance was considered when *p* < 0.05. All analyses and estimations were adjusted for the survey’s sampling design using the STATA SE v14 (Stata Corp LLC, TX, USA) [47] SVY module for complex samples.

## 3. Results

A total of 11,008 children and 8599 adolescents from Mexico were studied (Figure 1). In 2006 the total energy intake in children was 1538 kcal, and the contribution of energy intake by UPPs was 27.3%; in 2016, the total energy intake was 1606 kcal, and contributions of UPPs were 30.1%. The total energy intake in adolescents in 2006 was 1770 kcal, and the contribution of energy intake by UPPs was 23.0%, while in 2016, the total energy intake was 2023 kcal. The contributions of UPPs were 28.3% (Table 1).

Appendix A shows the sociodemographic characteristics stratified by socioeconomic status in children. The total energy intake was 1391 kcal/d in the lower tertile of SES in 2006, and the contribution of energy intake by UPPs was 19.4%, while in the highest tertile, the total energy intake was 1681 kcal/d and 34.4% by UPPs. In 2016, in the lowest tertile, total energy intake was 1524 kcal/d and 23.9% by UPPs, and the highest tertile was 1707 kcal/d and 35.6% by UPPs. Appendix A shows the sociodemographic characteristics stratified by socioeconomic status in adolescents. The total energy intake was 1684 kcal/d in the lower tertile of SES in 2006, and the contribution of energy intake by UPPs was 15.8%, while in the highest tertile, the total energy intake was 1846 kcal/d and 29% by UPPs. In 2016, in the lowest tertile, total energy intake was 1976 kcal/d and 23.7% by UPPs, and the highest tertile was 2080 kcal/d and 32% by UPPs.

Table 2 shows an association between excess weight and the double burden of malnutrition and consumption of UPPs in the lowest tertile of SES in children in model 1 and model 2. For each 200 kcal/d of increased consumption of UPPs, the odds of having excess weight were 16% higher, and the odds of being with DBM were 30% higher. In adolescents, we observed a positive association between UPPs and the DBM in the higher tertile of SES (26% higher odds). In the analysis stratified by year and SES (Table 3), the associations between consumption of UPPs and excess weight and DBM were significant in 2016 in children; however, in adolescents, the association between UPPs and DBM was observed in both years.

Table 4 shows a pseudo-panel analysis. The association of higher consumption of UPPs is differential. The results show a positive association between the consumption of UPPs and increased prevalence of DBM in the lowest tertile of SES, which represents low SES, and a negative association in tertile 2 of SES, which represents medium SES, to the prevalence of anemia.

## 4. Discussion

We evaluated the association between the coexistence of anemia and excess weight (overweight or obesity) at an individual level as DBM and the consumption of UPPs in children and adolescents. The consumption of UPPs was positively associated with the DBM and excess weight in the lowest SES in children, while in adolescents, the association between DBM and consumption of UPPs was in the higher SES.

The results of this research support the associations between the consumption of UPPs and excess weight in children reported in another study [27]. This association was consistently observed only in children from lower SES. The absence of clear associations in children and adolescents was reported previously by Lane et al. [48]. This absence is partly explained by physical and sedentary behavior and the underreporting of dietary intake [48]. In our study, we adjusted by screen time in adolescents; however, screen time is a tiny part of sedentary behavior, and we did not have information to include physical activity. Possible mechanisms to explain the associations between UPPs and excess weight are pathways of energy consumption, healthy food substitution, change in metabolic function, and critical nutrients. UPPs increase energy intake, which replaces healthy foods, alters the intestinal microbiome, and may lead to a more dysfunctional metabolism [20,49,50]. The UPPs were positively associated with nutrients related to chronic diseases (e.g., obesity), such as added sugar, total and saturated fat, and decreased protein and dietary fiber [20].

The results of this study did not show associations between the consumption of UPPs and anemia. Although a Mexican study recently reported that increased consumption of UPPs decreased the Mexican population’s dietary diversity and micronutrient intake (niacin, pantothenic acid, pyridoxine, folate, vitamin B_12_, vitamin C, vitamin E, zinc, calcium, magnesium, potassium, and phosphorus), no statistical differences were observed in intake of vitamins A, D, thiamine, riboflavin, and iron across quintiles of energy contribution of UPPs [18]. Deficiencies of iron, vitamins A, B_12_, C, D, and E, riboflavin, pyridoxine, folate, and copper can also result in anemia due to their specific roles in the production of hemoglobin or erythrocytes [51]. Furthermore, UPPs are frequently fortified [52]. The addition of micronutrients (as a persuasive marketing technique for incorporating health and nutritional claims) in UPPs might increase their consumption, giving them a “health halo effect” that leads consumers to overestimate their nutritional quality and not to avoid the risk of adverse effects in health, such as obesity, diabetes, high blood pressure, and other noncommunicable diseases [53].

Our results support that the consumption of UPPs was associated with excess weight and DBM. We show the association with higher consumption of UPPs and DBM in children with lower SES. The poor quality of nutrients in UPPs can contribute to the complex phenomenon of the DBM. A previous study has suggested that the co-occurrence of obesity with anemia could have common underlying factors, such as diets that are high in energy and deficient in micronutrients [54]. At the same time, another report showed the link between anemia and excess weight associated with metabolic changes that could influence micronutrient metabolism [55].

The consumption of UPPs in Mexico is higher in children and adolescents [17]. We showed that children were more vulnerable than adolescents to the DBM associated with more consumption of UPPs. Higher consumption of UPPs at younger ages is associated with the marketing and advertising of these products [56]. Children and adolescents are especially susceptible to advertising because they do not have the neurological maturity to understand the marketing intentions and to generate brand awareness, preference, and loyalty that persist through adulthood [57]. The companies that produce UPPs use persuasive advertising strategies for specific population targets, such as children and adolescents [58].

We observed that the consumption of UPPs was higher in 2016 than in 2006 in both age-groups, which coincides with the results of the Pan American Health Organization (PAHO) on the increase in the sales of these products in Mexico [59]. This study’s results support that reducing the consumption of UPPs is one of the double-duty actions that can tackle multiple forms of malnutrition [60,61].

Malnutrition has unfavorable consequences in the short and long term. Anemia has adverse effects on cognitive development, growth, and resistance to infection [4,6,62]. Moreover, overweight or obesity at younger ages leads to excess weight in adulthood and the early development of chronic diseases and mental health problems related to self-esteem, discrimination, and bullying [4,5]. In the long term, it affects their productivity and income [4]. Children and adolescents have intraindividual, interindividual, social, and environmental characteristics that leave them more vulnerable to the food system that influences their dietary practices [63]. On the other hand, children and adolescents are age-groups critical to adopting healthy diets.

### Strengths and Limits of the Study

In the present study, some limitations must be considered. First, the coexistence of anemia and excess weight as a definition of the DBM is limited. Other forms of malnutrition, such as diet-related noncommunicable diseases and micronutrient deficiencies at the individual, household, and population levels, are essential to monitor in terms of public health [8]. Therefore, this study only describes one component of the DBM at the individual level and does not consider the critical aspect of a lifetime [64]. Second, it is not possible to determine causality between UPPs and the coexistence of anemia and excess weight due to the nature of the data. However, we explored the pseudo-panel analysis to approximate the effect of consumption of UPPs across 10 years with health conditions. Third, the manufacturing of UPPs changes rapidly [10]. In Mexico, one of the particularly important foods is the corn tortilla. In this study, corn tortilla was considered an MPF. However, the industrialization of products that were initially being minimally processed is increasingly common. It is necessary to monitor foods of great cultural importance in Mexico, such as corn tortillas, and identify their contribution as UPPs to the population. On the other hand, Mexico adopted food warning labels in 2019 [65,66], and the reformulations of products are expected, as happened in Chile [67]; thus, more different forms of malnutrition can be monitored.

This study’s strengths are that first, we used nationally representative surveys in the analysis. Second, the SFFQ used as an instrument to assess diet was previously validated to identify UPPs [40].

## 5. Conclusions

In conclusion, the consumption of UPPs in Mexico increased among children and adolescents from 2006 to 2016. Our findings support the role of UPP consumption in excess weight in children and adolescents and the coexistence of DBM in children. This result was differentiated by socioeconomic status. Moreover, the DBM affects children with lower socioeconomic status. The data reinforce the need to regulate UPPs and show that the need to take action in the food environment should prioritize more vulnerable areas to protect the diet and health of the younger ages. This assessment supports the public policy initiatives being proposed in the country to regulate these products (front-of-pack labels and excise taxes on sugar-sweetened beverages and junk food).

## Figures and Tables

**Figure 1 nutrients-14-04347-f001:**
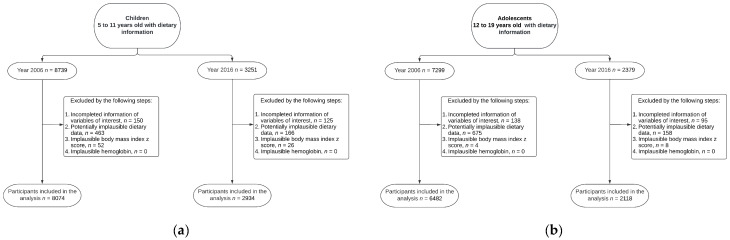
Participant flowchart. (**a**) Children, years 2006 and 2016. (**b**) Adolescents, years 2006 and 2016.

**Table 1 nutrients-14-04347-t001:** Characteristics of children and adolescents in 2006 and 2016.

	Children		Adolescents	
	Year 2006	Year 2016	Year 2006	Year 2016
Sample ^a^, *n*	8074	2934	6482	2118
Total energy intake kcal/d, mean (95% CI)	1538 (1513, 1563)	1606 (1563, 1649)	1770 (1742, 1799)	2023 (1966, 2080)
Ultraprocessed products				
kcal/d, mean (95% CI)	441 (424, 457)	499 (469, 528)	412 (397, 427)	590 (556, 624)
% of energy, mean (95% CI)	27.3 (26.5, 28.2)	30.1 (28.7, 31.6)	23.0 (22.2, 23.7)	28.3 (27.1, 29.5)
Age, mean (95% CI)	8.8 (8.7, 8.8)	8.0 (7.9, 8.1)	15.1 (15.1, 15.2)	15.3 (15.2, 15.5)
Sex, % (95% CI)				
Men	49.7 (47.9, 51.5)	51.4 (48.2, 54.6)	49.5 (47.5, 51.4)	49.8 (46.5, 53.1)
Women	50.3 (48.5, 52.1)	48.6 (45.4, 51.8)	50.5 (48.6, 52.5)	50.2 (46.9, 53.5)
Screen time, % (95% CI)				
less than 7 h/week	66.1 (62.6, 69.5)	6.7 (4.6, 9.6)	63.9 (61.9, 65.8)	5.4 (4.1, 7)
7 to 13 h/week	29.6 (26.5, 32.9)	13.6 (9.7, 18.7)	29.8 (28.1, 31.6)	11.7 (9.7, 14)
14 to 20 h/week	3.9 (2.3, 6.4)	14.6 (10.8, 19.5)	5.1 (4.3, 6.1)	11.3 (9.1, 13.8)
21 or more h/week	0.4 (0.1, 1.3)	65.1 (58.4, 71.3)	1.1 (0.6, 2.1)	71.7 (68.3, 74.8)
Head of household educational level ^b^				
Low	62.7 (60.3, 65)	38.1 (34.1, 42.2)	67.7 (65.6, 69.8)	45.9 (42.1, 49.8)
Medium	33 (30.8, 35.2)	53.8 (49.7, 57.8)	28.3 (26.3, 30.3)	46.8 (43, 50.7)
High	4.3 (3, 6)	8.2 (5.1, 12.8)	4 (3.1, 5.1)	7.2 (5.1, 10.3)
Ethnicity				
Indigenous	23.2 (20.8, 25.9)	4.2 (2.6, 6.6)	23.4 (20.9, 26)	4.7 (3, 7.2)
Nonindigenous	76.8 (74.1, 79.2)	95.8 (93.4, 97.4)	76.6 (74, 79.1)	95.3 (92.8, 97)
Socioeconomic status ^c^, % (95% CI)				
Tertile 1	35.4 (32.6, 38.2)	34 (29.6, 38.8)	32.6 (29.9, 35.4)	32.7 (29.2, 36.3)
Tertile 2	32.7 (30.6, 34.8)	34.5 (30.2, 39)	33.9 (31.8, 36.1)	33.6 (30.2, 37.2)
Tertile 3	32 (29.4, 34.6)	31.5 (26.7, 36.7)	33.5 (31.1, 35.9)	33.7 (29.8, 37.9)
Area ^d^, % (95% CI)				
Urban	61.3 (58.1, 64.4)	71.2 (66.9, 75.2)	58.7 (55.8, 61.6)	71.7 (68.2, 74.9)
Rural	38.7 (35.6, 41.9)	28.8 (24.8, 33.1)	41.3 (38.4, 44.2)	28.3 (25.1, 31.8)
Region, % (95% CI)				
North	12.6 (10.5, 15.1)	20.5 (15.9, 26.1)	13.4 (11.5, 15.6)	17.1 (13.8, 20.9)
Central and Mexico City	49.9 (45.4, 54.4)	47 (41.6, 52.4)	47.5 (43.3, 51.8)	49.2 (44.9, 53.4)
South	37.5 (33.4, 41.8)	32.5 (27.8, 37.5)	39.1 (35.1, 43.2)	33.8 (29.5, 38.3)
Health conditions				
Anemia ^e^	12.1 (10.8, 13.6)	12.3 (10.6, 14.3)	8.5 (7.5, 9.6)	10.6 (8.8, 12.8)
Excess weight ^f^	31.5 (29.5, 33.6)	32.7 (28.9, 36.8)	30.4 (28.7, 32.1)	36 (32.8, 39.3)
Double Burden of Malnutrition ^g^	2.7 (2.2, 3.3)	3.2 (2.3, 4.4)	2.8 (2.2, 3.6)	2.9 (2.1, 4)

^a^ Estimated population (thousand): children—2006, *N* = 9357.5; 2016, *N* = 15,777.6; Adolescents—2006, *N* = 9126.9; 2016, *N* = 15,449.7. ^b^ Head of household educational level: low—no formal education to elementary school; medium—middle and high school education; and high—college and postgraduate courses. ^c^ Socioeconomic status was derived using the principal component method; the index score was divided into tertiles to represent low, medium, and high socioeconomic status. ^d^ Area rural defined by 1. Rural localities with fewer than 2500 inhabitants and urban with 2500 or more. ^e^ Anemia: Hb concentrations to diagnose anemia adjusted by altitude were <11.5 g/dL for children 5 to 11 years old; <12.0 g/dL for adolescents 12 to 14 years old; <12.0 g/dL for females ≥15 years old, and <13.0 g/dL for males ≥15 years old. ^f^ Excess weight: overweight or obesity. Children and adolescents with z scores above +1 were classified as overweight, and those with z scores above +2 standard deviations as obese. ^g^ Individuals with anemia and overweight or obesity.

**Table 2 nutrients-14-04347-t002:** Adjusted analyses of the association between the dietary intake of ultraprocessed products and health conditions in Mexican children and adolescents.

	General Model		Socioeconomic Status					
			Tertile 1		Tertile 2		Tertile 3	
	Model 1 ^a^	Model 2 ^b^	Model 1 ^a^	Model 2 ^b^	Model 1 ^a^	Model 2 ^b^	Model 1 ^a^	Model 2 ^b^
Children	OR, (95% CI)	OR, (95% CI)	OR, (95% CI)	OR, (95% CI)	OR, (95% CI)	OR, (95% CI)	OR, (95% CI)	OR, (95% CI)
Anemia ^c^								
No	1.0	1.0	1.0	1.0	1.0	1.0	1.0	1.0
Yes	0.93, (0.86, 1.01)	0.97, (0.88, 1.06)	1.03, (0.93, 1.15)	1.03, (0.92, 1.16)	0.91, (0.79, 1.06)	0.93, (0.79, 1.09)	0.94, (0.81, 1.08)	0.96, (0.83, 1.11)
Excess weight ^d^								
No		1.0	1.0	1.0	1.0	1.0	1.0	1.0
Yes	1.13, (1.08, 1.2)	1.09, (1.02, 1.16)	1.2, (1.11, 1.3)	1.16, (1.07, 1.26)	0.99, (0.92, 1.06)	1.01, (0.94, 1.09)	1.08, (0.96, 1.21)	1.09, (0.96, 1.24)
Double Burden of Malnutrition ^e^								
No	1.0	1.0	1.0	1.0	1.0	1.0	1.0	1.0
Yes	1.07, (0.96, 1.18)	1.12, (0.99, 1.26)	1.37, (1.15, 1.63)	1.3, (1.07, 1.58)	0.97, (0.8, 1.18)	1.04, (0.88, 1.21)	1, (0.83, 1.2)	1.07, (0.88, 1.29)
Adolescents								
Anemia ^c^								
No	1.0	1.0	1.0	1.0	1.0	1.0	1.0	1.0
Yes	0.95, (0.87, 1.04)	0.99, (0.89, 1.1)	1.01, (0.85, 1.2)	1.04, (0.86, 1.26)	0.89, (0.76, 1.04)	0.91, (0.77, 1.09)	1.05, (0.9, 1.23)	1.09, (0.92, 1.28)
Excess weight ^d^								
No	1	1.0	1.0	1.0	1.0	1.0	1.0	1.0
Yes	1.06, (1.01, 1.12)	1.04, (0.98, 1.1)	1.06, (0.96, 1.17)	1.02, (0.93, 1.13)	1.04, (0.96, 1.14)	1.03, (0.94, 1.13)	1.04, (0.94, 1.14)	1.03, (0.94, 1.14)
Double Burden of Malnutrition ^e^								
No	1	1.0	1.0	1.0	1.0	1.0	1.0	1.0
Yes	1.09, (0.95, 1.25)	1.17, (1.02, 1.35)	1.17, (0.87, 1.58)	1.29, (0.97, 1.71)	0.94, (0.74, 1.21)	1.03, (0.81, 1.31)	1.25, (1.02, 1.54)	1.26, (1.03, 1.55)

^a^ Adjusted by year and energy NOVA groups. ^b^ Children: adjusted variables model 1 + age quintiles, socioeconomic status, ethnicity, area, and region. Adolescents: adjusted variables model 1 + age quintiles, socioeconomic status, ethnicity, area, region screen time, and sex. ^c^ Anemia: Hb concentrations to diagnose anemia adjusted by altitude were <11.5 g/dL for children 5 to 11 years old, <12.0 g/dL for adolescents 12 to 14 years old, <12.0 g/dL for females ≥15 years old, and <13.0 g/dL for males ≥15 years old. ^d^ Excess weight: overweight or obesity. Children and adolescents with z scores above +1 were classified as overweight, and those with z scores above +2 standard deviations as obese. ^e^ Individuals with anemia and overweight or obesity.

**Table 3 nutrients-14-04347-t003:** Adjusted analyses of the association between the dietary intake of ultraprocessed products and health conditions in Mexican children and adolescents by year.

	General Model ^a^		Socioeconomic Status ^b^					
			Tertile 1		Tertile 2		Tertile 3	
	2006	2016	2006	2016	2006	2016	2006	2016
Children	OR, (95% CI)	OR, (95% CI)	OR, (95% CI)	OR, (95% CI)	OR, (95% CI)	OR, (95% CI)	OR, (95% CI)	OR, (95% CI)
Anemia ^c^								
No	1.0	1.0	1.0	1.0	1.0	1.0	1.0	1.0
Yes	0.98, (0.9, 1.06)	0.97, (0.85, 1.11)	0.94, (0.84, 1.05)	1.15, (0.96, 1.37)	1.03, (0.91, 1.16)	0.87, (0.68, 1.12)	1, (0.85, 1.19)	0.93, (0.77, 1.13)
Excess weight ^d^								
No		1.0	1.0	1.0	1.0	1.0	1.0	1.0
Yes	1.02, (0.96, 1.07)	1.13, (1.03, 1.24)	1.07, (0.99, 1.15)	1.29, (1.11, 1.51)	0.97, (0.9, 1.05)	1.04, (0.93, 1.16)	1, (0.89, 1.12)	1.13, (0.96, 1.33)
Double Burden of Malnutrition ^e^								
No	1.0	1.0	1.0	1.0	1.0	1.0	1.0	1.0
Yes	1.15, (1.04, 1.29)	1.08, (0.9, 1.3)	1.1, (0.9, 1.34)	1.47, (1.1, 1.96)	1.2, (1.04, 1.39)	0.87, (0.66, 1.14)	1.14, (0.89, 1.45)	1.05, (0.81, 1.36)
Adolescents								
Anemia ^c^								
No	1.0	1.0	1.0	1.0	1.0	1.0	1.0	1.0
Yes	1.09, (0.99, 1.19)	0.97, (0.84, 1.11)	0.95, (0.82, 1.11)	1.12, (0.84, 1.49)	1.12, (0.96, 1.32)	0.86, (0.69, 1.09)	1.19, (1.04, 1.37)	1.05, (0.84, 1.31)
Excess weight ^d^								
No	1	1.0	1.0	1.0	1.0	1.0	1.0	1.0
Yes	1.04, (0.98, 1.1)	1.04, (0.97, 1.12)	1.02, (0.92, 1.13)	1.04, (0.88, 1.22)	1.08, (0.98, 1.18)	1.01, (0.9, 1.13)	0.96, (0.88, 1.06)	1.03, (0.91, 1.16)
Double Burden of Malnutrition ^e^								
No	1	1.0	1.0	1.0	1.0	1.0	1.0	1.0
Yes	1.08, (0.93, 1.24)	1.21, (1.02, 1.44)	0.99, (0.79, 1.23)	1.6, (1.13, 2.26)	1.13, (0.86, 1.49)	0.98, (0.67, 1.42)	1, (0.81, 1.22)	1.27, (0.97, 1.67)

^a^ Children: adjusted by year, energy NOVA groups, age quintiles, socioeconomic status, ethnicity, area, and region. Adolescents: adjusted by year, energy NOVA groups, age quintiles, socioeconomic status, ethnicity, area, region screen time, and sex. ^b^ Socioeconomic status was derived using the principal component method; the index score was divided into tertiles to represent low, medium, and high socioeconomic status. ^c^ Anemia: Hb concentrations to diagnose anemia adjusted by altitude were <11.5 g/dL for children 5 to 11 years old, <12.0 g/dL for adolescents 12 to 14 years old, <12.0 g/dL for females ≥15 years old, and <13.0 g/dL for males ≥15 years old. ^d^ Excess weight: overweight or obesity. Children and adolescents with z scores above +1 were classified as overweight, and those with z scores above +2 standard deviations as obese. ^e^ Individuals with anemia and overweight or obesity.

**Table 4 nutrients-14-04347-t004:** Pseudo-panel analysis ^a^ of the association between the dietary intake of ultraprocessed products and health conditions in a cohort of Mexican children.

	Socioeconomic Status ^b^					
	Tertile 1		Tertile 2		Tertile 3	
Model ^c^	β, (95% CI)	*p*	β, (95% CI)	*p*	β, (95% CI)	*p*
Anemia ^d^	0.036, (−0.005, 0.078)	0.079	−0.148, (−0.263, −0.034)	0.017	−0.065, (−0.133, 0.003)	0.059
Excess weight ^e^	−0.017, (−0.1, 0.066)	0.652	−0.06, (−0.13, 0.01)	0.086	0.03, (−0.1, 0.16)	0.588
Double Burden of Malnutrition ^f^	0.048, (0.029, 0.067)	<0.001	−0.059, (−0.182, 0.065)	0.306	−0.009, (−0.052, 0.034)	0.633

^a^ The pseudo-panel analysis was constructed cells restricted the birth cohort from 1997 to 2001 for every country region, place of residence, socioeconomic status, and sex. ^b^ Socioeconomic status was derived using the principal component method; the index score was divided into tertiles to represent low, medium, and high socioeconomic status. ^c^ Model: stratified analysis by socioeconomic status, adjusted models by the delta of no processed or minimally processed foods and health conditions in 2006. ^d^ Anemia: Hb concentrations to diagnose anemia adjusted by altitude were <11.5 g/dL for children 5 to 11 years old, <12.0 g/dL for adolescents 12 to 14 years old, <12.0 g/dL for females ≥15 years old, and <13.0 g/dL for males ≥15 years old. ^e^ Excess weight: overweight or obesity. Children and adolescents with z scores above +1 were classified as overweight, and those with z scores above +2 standard deviations as obese. ^f^ Individuals with anemia and overweight or obesity.

## Data Availability

Data described in the manuscript and codebook are made publicly and freely available without restriction at https://ensanut.insp.mx/ (accessed on 22 August 2022). The analytic code will be made available upon request to the corresponding author EAMF, eric@insp.mx.

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
