# Peer review of "Trend of Ultraprocessed Product Intake Is Associated with the Double Burden of Malnutrition in Mexican Children and Adolescents"

_nutrients, 2022, doi:10.3390/nu14204347_

Round 1

Reviewer 1 Report

Thanks for providing the opportunity to review the article on a very important issue. Overall it is good but a few suggestions seem to be helpful

English language issues in some places could be rechecked

Title: every first word must be uppercase as

Trend of Ultra-processed Products Intake is Associated with the Double Burden of Malnutrition in Mexican Children and Adolescents

Abstract: the first sentence in the abstract needs to be revised and made more meaningful.

Although there is evidence that ultra-processed products (UPPs) have a poor nutritional profile, it is unclear whether there is an association between UPPs and the double burden of malnutrition (DBM) at the individual level. 

Results: line 198 must be rewritten as it is survey data. replaced the word participated here with a suitable word choice; it might be written as:

A total of 11008 children and 8599 adolescents from Mexico were studied.

Discussion: Give a heading of the strength and limits of study at the end of the discussion section

The discussion section needs to provide a few more citations/literature; better references used in the intro must not be excessively repeated unless unavoidable

Conclusion: is too short and sentences must be broken down into two (see lines 338-342)

Author Response

Point 1: Thanks for providing the opportunity to review the article on a very important issue. Overall it is good but a few suggestions seem to be helpful. English language issues in some places could be rechecked.

Response 1: Thank you for this observation. The manuscript was reviewed to improve the English language.

Point 2: Title: every first word must be uppercase as

Trend of Ultra-processed Products Intake is Associated with the Double Burden of Malnutrition in Mexican Children and Adolescents

Response 2: We have changed the title as suggested. Thank you for this observation.

Point 3: Abstract: the first sentence in the abstract needs to be revised and made more meaningful.

Although there is evidence that ultra-processed products (UPPs) have a poor nutritional profile, it is unclear whether there is an association between UPPs and the double burden of malnutrition (DBM) at the individual level. 

Response 3: We appreciate your comment. Please see changes accordingly in lines 14-16.

The ultra-processed products (UPPs) have been associated with unfavorable health outcomes; however, until now, they have not been associated with the coexistence of undernutrition and overnutrition, known as the double burden of malnutrition (DBM) at the individual level.

Point 4: Results: line 198 must be rewritten as it is survey data. replaced the word participated here with a suitable word choice; it might be written as:

A total of 11008 children and 8599 adolescents from Mexico were studied.

Response 4: Thank you for this suggestion; we have changed the phrasing as suggested in line 199.

Point 5: Discussion: Give a heading of the strength and limits of study at the end of the discussion section

Response 5: Done. Please see changes accordingly in line 308.

Point 6: The discussion section needs to provide a few more citations/literature; better references used in the intro must not be excessively repeated unless unavoidable

Response 6: Thank you for this observation. The manuscript was reviewed and add few more citations in the discussion section on lines 314 and 319. The authors considered it necessary to repeat some cites in the document.

Point 7: Conclusion: is too short and sentences must be broken down into two (see lines 338-342)

In conclusion, our findings support the role of UPPs consumption in an excess weight in children and adolescents and the coexistence of DBM in children. The data reinforces the need for regulated UPPs to protect the diet and health of the younger ages; and it supports the public policy initiatives that are being proposed in the country for the regulation of these products (front-of-pack label and sugar-sweetened beverages and junk-food excise taxes).

Response 7: We appreciate your comment. Please see changes accordingly in lines 331-339.

In conclusion, the consumption of UPPs in Mexico increased among children and adolescents from 2006 to 2016. Our findings support the role of UPPs consumption in excess weight in children and adolescents and the coexistence of DBM in children. This result was differentiated by socioeconomic status. Moreover, the DBM affects children with lower socioeconomic status. The data reinforces the need to regulate UPPs and shows that the need to take action in the food environment should prioritize areas more vulnerable to protect the diet and health of the younger ages. This assessment supports the public policy initiatives being proposed in the country to regulate these products (front-of-pack labels and excise taxes on sugar-sweetened beverages and junk food).

Reviewer 2 Report

This study confirm the important findings supporting the role of UPPs consumption in an excess weight  in children and adolescents and the coexistence of DBM in children.

These data reinforces the need for regulated UPPs to protect the diet and health of the younger ages

 These findings supports the public policy initiatives that are being proposed in the country for the regulation of these products (front-of-pack label and sugar-sweetened beverages and junk- food excise taxes).

Author Response

Point 1: This study confirm the important findings supporting the role of UPPs consumption in an excess weight in children and adolescents and the coexistence of DBM in children.

Response 1: We appreciate this comment received on the article.

Point 2: These data reinforces the need for regulated UPPs to protect the diet and health of the younger ages

Response 2: We appreciate this comment received on the article.

Point 3: These findings supports the public policy initiatives that are being proposed in the country for the regulation of these products (front-of-pack label and sugar-sweetened beverages and junk- food excise taxes).

Response 3: We appreciate this comment received on the article.